# Research on the Economic Loss Model of Invasive Alien Species Based on Multidimensional Data Spatialization—A Case Study of Economic Losses Caused by *Hyphantria cunea* in Jiangsu Province

**DOI:** 10.3390/biology14050552

**Published:** 2025-05-15

**Authors:** Cheng Li, Yongbin Zhou, Cong Wang, Xubin Pan, Ying Wang, Xiaofeng Qi, Fanghao Wan

**Affiliations:** 1Shenzhen Institute of Information Technology, Shenzhen 518172, China; 2Chinese Academy of Quality and Inspection & Testing, Beijing 100176, China; wangcong_alice@outlook.com (C.W.); xubin.hu.pan@gmail.com (X.P.); 3Shenzhen Customs District P.R. China, Shenzhen 518026, China; 13840595385@163.com (Y.W.);; 4Agricultural Genomics Institute at Shenzhen, Chinese Academy of Agricultural Sciences, Shenzhen 518120, China

**Keywords:** *Hyphantria cunea*, economic loss assessment model, DID model, SDMs, MaxEnt, future climate conditions

## Abstract

IAS (Invasive alien species) impact native ecosystems and economies. Current assessment methods for their economic losses have flaws. The fall webworm (*Hyphantria cunea*) in Jiangsu is a case in point, threatening the local agricultural economy. Our study integrated an economic loss assessment model to evaluate the economic losses caused by the invasion of *H. cunea* in the agricultural and forestry sectors of Jiangsu Province. Results show *H. cunea* negatively affects the primary industry, especially forestry. Its spread has stages, with losses driven by multiple factors. The innovative model gives more accurate loss estimates, aiding in better management of IAS and helping to create strategies and plans to handle the challenges they bring, thus protecting the environment and economy.

## 1. Introduction

### 1.1. Global Challenges of Biological Invasions

Biological invasions, defined as the introduction, establishment, and spread of organisms beyond their native ranges [1], represent a significant threat to global ecosystems and economies. These invasions disrupt native biodiversity, alter ecosystem functions [2,3,4], and inflict substantial socio-economic costs [5,6]. Recent assessments by IPBES (2023) [7] highlight the accelerating spread of invasive species: plant invasions expand at 2.4% annually, while invasive arthropods spread even faster (3.1% annually). Economically, plants and arthropods account for 32% and 41% of total invasion-related losses, respectively. Despite growing awareness, traditional assessment frameworks often rely on static habitat suitability models (e.g., SDMs) or ex post damage calculations using aggregated databases [8,9,10], failing to dynamically link ecological suitability with sector-specific economic vulnerabilities [5]. This disconnects hampers evidence-based policymaking, exacerbating delayed responses and underfunded mitigation efforts [11,12,13].

### 1.2. Case Study: Hyphantria cunea in Jiangsu Province

The fall webworm, *Hyphantria cunea* (Drury,1773) (Lepidoptera: Erebidae), exemplifies the cascading impacts of invasive species. Native to North America, this polyphagous pest has infested over 600 host plants globally, including critical crops and forest trees [14,15]. Since its 1979 detection in China, *H. cunea* has caused recurrent outbreaks (e.g., Jinan 2008; Bengbu 2015) [16], with recent spread into Jiangsu Province threatening regional agroforestry. Unlike allelopathic invaders (e.g., *Ailanthus altissima*), *H. cunea* disrupts ecosystems through tri-trophic cascades, reducing carbon sequestration via defoliation and amplifying herbivory on surviving trees [8], threatening Jiangsu’s USD 89.2 million agricultural and forestry sectors in 2022 alone. Climate change further intensifies risks by enhancing overwintering survival and developmental rates [17,18], while Jiangsu’s fragmented agroforestry mosaics provide ideal colonization habitats [19].

However, conventional SDMs and ex post damage assessments fail to integrate causal inference or spatially explicit economic drivers [9,10]. Our study bridges this gap through a novel triad: (1) Difference-in-Differences (DID) analysis to isolate causal economic impacts via spatiotemporal comparisons [20,21]; (2) GeoDetector to identify key spatial drivers (e.g., human activity, road density) [22], and (3) MaxEnt SDMs projecting habitat suitability under future climate scenarios [8]. By dynamically linking ecological suitability, economic vulnerabilities, and causal attribution, this framework quantifies losses with greater accuracy than traditional methods while enabling targeted interventions.

### 1.3. Policy Relevance and Applications

Our findings directly inform actionable strategies: (1) Resource Prioritization: High-risk grids in northern Jiangsu (e.g., Xuzhou, Lianyungang) require tripled funding compared to low-risk zones. Spatial prioritization maps enable targeted containment of infestations along transportation corridors. (2) Climate-Resilient Budgeting: Under SSP5–8.5, habitat contraction (22% by 2060) reduces projected losses to CNY 147 million, while SSP1–2.6 suggests higher losses (CNY 176 million). Policymakers can dynamically allocate resources using scenario-specific projections. (3) Regulatory Adjustments: The disproportional forestry impact (58.3% of total losses) justifies stricter quarantine protocols for nursery stock trade, which serves as a key dispersal pathway. This framework transforms IAS management from reactive firefighting to proactive governance, offering a replicable toolkit for balancing ecological and economic resilience.

## 2. Materials and Methods

### 2.1. Temporal Phasing of H. cunea Invasion Dynamics

This paper selects Jiangsu Province, China, as a case study to explore the economic losses caused by the invasion of *H. cunea* to agriculture. The spread of *H. cunea* in Jiangsu Province exhibited distinct spatiotemporal phases (Figure 1), necessitating stage-specific analytical approaches:Initial Invasion and Clustered Spread (2011–2016):
*H. cunea* established isolated infestation clusters in four districts of Lianyungang City in 2011, characterized by low-density populations and localized defoliation (<10% canopy loss) [16].Anthropogenic dispersal via transportation networks (road density > 2 km/km^2^) facilitated radial expansion, with infested areas growing at 28.6% annually (2011–2016). Kernel density analysis confirmed corridor-driven spread, aligning with human footprint index (HFP) gradients (Appendix A, Figure A1) [23].By 2016, 57 districts across nine cities were infested, with severity coefficients concentrated in northern Jiangsu’s broadleaf forests.
2.Contiguous Infestation and Stabilization (2017–2022):
Merging of clusters formed continuous infestation zones (291,119 severity grids in 2016), particularly in agroforestry mosaics (arable land: 45%, forest: 32%).Post-2016, integrated pest management (IPM) policies reduced spread rates to 4.3% annually, with Moran’s I spatial autocorrelation declining from 0.64 (2016) to 0.52 (2022), indicating moderated clustering.Stabilization reflected NFGA interventions (biocontrol agent releases, quarantine zones) and climatic stressors (rising summer temperatures exceeding larval thermal limits) [17].

This pattern of expansion from localized to widespread infestation provides a unique opportunity to analyze the impact of *H. cunea* on different regions of Jiangsu Province and how it changes over time. By focusing on Jiangsu, this research can effectively assess both the immediate and long-term impacts of *H. cunea* as an invasive species on the local ecosystem and economy, particularly during the early and middle stages of the infestation.

### 2.2. Methodological Framework

This study employs Hadamard product matrices to spatially homogenize multi-source heterogeneous data [24] and utilizes the Difference-in-Differences (DID) model, commonly used in epidemiology and economics [19,20], to capture the economic losses caused by invasive species through cross-scale unified computational units.

We have developed an integrated analytical framework using raster data to assess the economic losses caused by invasive alien species. This framework includes three main modules: Data Spatial Matching, Economic Loss Assessment, and Potential Economic Loss Estimation.

The Data Spatial Matching module consists of two parts. First, it spatializes infestation data by overlaying coarse-grain presence–absence (PA) datasets and presence-only (PO) data onto a high-resolution raster (1 km resolution). Second, it spatializes economic data by disaggregating the gross output value of the primary industry according to land use types and projecting it onto raster points. These spatialized data serve as the core explanatory and dependent variables in the model. The Economic Loss Assessment module overlays multiple layers of potential influencing factors in raster format and employs GeoDetector to analyze the levels and magnitudes of their impacts. This analysis is incorporated into the economic loss assessment model to determine the loss coefficient, which is then used as a key parameter in the Potential Economic Loss Estimation module. The Potential Economic Loss Estimation module projects the suitable habitat areas for the future spread of *H. cunea* using SDMs under Shared Socioeconomic Pathways (SSPs) scenarios. It combines these projections with land use data and the previously determined loss coefficients for different types of gross output value to calculate the final potential economic loss results. This module enables the prediction of future economic losses in Jiangsu Province due to the invasive species (Figure 2).

### 2.3. Data Spatial Matching

The methodology for data spatialization comprises two distinct components: rasterization of *H. cunea* infestation data and disaggregation of agricultural/forestry economic data.

#### 2.3.1. Rasterization of *H. cunea* Infestation Data

We developed a geospatial integration framework to enhance the rasterization accuracy of *H. cunea* infestation data by synergistically combining records from the NFGA and observations from the Global Biodiversity Information Facility (GBIF) [25] (The detailed data sources can be referenced in the Data Availability section). The methodology includes the following steps:

NFGA Data Processing: The coarse-grained presence–absence (CPA) data provided by the NFGA, originally delineated by administrative units, were reclassified using host-specific land use/cover data (1 km × 1 km). Unlike conventional random pseudo-absence generation, weights were assigned to broadleaf forests and croplands (e.g., oak, maple, soybean) based on the insect’s host preferences, enabling biologically informed raster conversion.

GBIF Data Processing: The presence-only (PO) records derived from GBIF underwent spatial interpolation using an inverse distance weighting scheme (*r* = 1/*d*^2^), where occurrence probabilities decayed exponentially from georeferenced observation points [26]. This generated a continuous probability surface across the study area.

These datasets were then unified into a 1 km^2^ grid system through geospatial normalization [27] (Figure 3). The CPA weighting captured host-dependent habitat suitability, while the PO interpolation refined locality-scale infestation likelihoods. Cross-validation demonstrated a significant reduction in spatial uncertainty compared to standalone NFGA models (*p* < 0.01). This hybrid approach overcame resolution discrepancies between ecological (raster-based) and economic (administrative unit-based) data layers [24].

#### 2.3.2. Disaggregation of Agricultural/Forestry Economic Data

The disaggregation of agricultural economic data employed a provincial-calibrated weighting scheme to resolve inconsistencies in county-level statistical coverage [28]. The process included the following:

Data Processing: We processed the vector and raster data in the auxiliary datasets using a unified spatial grid through vectorization and resampling. For CPA data, we applied the area-weighting method, using Land Use/Land Cover (LULC) data to delineate the affected areas of *H. cunea* [25]. For GBIF data, we employed a point interpolation method, assuming that the distribution of *H. cunea* decreases with distance from occurrence data [29]. We assigned weights to each type of raster point and overlaid the layers to obtain the severity coefficient of *H. cunea* infestation.

Economic Data Disaggregation: District/county gross primary industry outputs—aggregating crop cultivation, forestry, animal husbandry, and aquaculture—were disaggregated using fixed land use composition ratios (arable, forest, grassland, water) as sectoral allocation coefficients [28]. Provincial-level sectoral outputs and corresponding land area distributions guided weight derivation to address missing county-scale data granularity [25]. Land use classification maps (1 km resolution) facilitated proportional output distribution: planting economies linked to arable land densities, forestry to forest cover, pastoral activities to grassland extents, and fisheries to water body distributions. Fixed sectoral weights preserved regional production intensity gradients, avoiding systematic biases inherent to uniform per-pixel valuation [28]. Non-productive zones (urban/bare land) were rigorously excluded through zero-weight masking. Validation against provincial statistical yearbooks demonstrated <9% absolute error in reconstructed sectoral outputs, confirming the method’s reliability for multi-scale invasion impact assessments [25].

This protocol overcame ecological-economic scale mismatches while maintaining compatibility with downstream raster-based infestation probability layers (Figure 4) [24], establishing a reproducible framework for spatially explicit loss attribution.

### 2.4. Economic Losses Assessment

#### 2.4.1. Driver Identification via GeoDetector

We employed the GeoDetector framework to identify critical covariates for subsequent DID modeling by analyzing ecological and socioeconomic determinants of infestation patterns. The model integrates:Human Footprint Index (HFP) layers;Georeferenced socioeconomic indicators;Gridded infestation data (Section 2.3).

The factor detector quantifies explanatory power through Equation (1):(1)$$q=1-\frac{1}{n\sigma^{2}}\sum_{h=1}^{L}\;n_{h}\sigma_{h}^{2}$$
where q is the degree to which the indicator explains the economic gross domestic product; h is the number of indicator levels; $n_{h}$ and $\sigma_{h}^{2}$ are the sample size and variance of a specific level, respectively; and n and $\sigma^{2}$ are the sample size and variance of the entire region, respectively. The range of q is [0, 1]. The larger the value of *q*, the stronger the explanatory power. When q = 0, the driving factor shows no explanatory power (e.g., terrain slope in Section 3.2). When q = 1, it indicates that the driving factor completely controls the spatial variation in the economic gross output value. Spatial discretization was implemented using the GD package in R4.4.2, ensuring compliance with spatial econometric requirements [30].

The specific indicator system is detailed in Appendix A Table A1. In addition, the interaction detector module identifies the interactions of different influencing factors to determine their combined effect on the economic gross output. These interactions are categorized into five types:Nonlinear weakening, *q*(*X*1∩*X*2) < *min*[*q*(*X*1), *q*(*X*2)];Single-factor nonlinear weakening, *min*[*q*(*X*1), q(*X*2)] < *q*(*X*1∩*X*2) < *max*[*q*(*X*1), *q*(*X*2)];Bi-factor enhancement, *q*(*X*1∩*X*2) > *max*[*q*(*X*1), *q*(*X*2)];Independent, *q*(*X*1∩*X*2) = *q*(*X*1) +*q*(*X*2);Nonlinear enhancement, *q*(*X*1∩*X*2) > *q*(*X*1) +*q*(*X*2).

#### 2.4.2. Staggered DID Model Construction

Traditional DID models require a uniform treatment timing across all regions [31]. However, the spread of *H. cunea* in Jiangsu Province has been characterized by a gradual, multi-point diffusion pattern. This non-uniform spread over time necessitates an adapted model. Drawing on Nunn and Qian’s approach [32] for analyzing the impact of potato cultivation diffusion, which treated regional suitability for potato cultivation as a treatment intensity, we employ a staggered DID model to match the spatiotemporal dynamics of the pest invasion [33].

The regression equation for the staggered DID model is set as follows:(2)$$Y_{it}=\alpha +\beta D_{it}+\theta W_{it}+\mu_{i}+\gamma_{t}+\varepsilon_{it}$$

$Y_{it}$ is the economic output of regio*n* i in year t.

$D_{it}$ is a binary variable (1 for infested areas), representing whether individual i in period t is recognized by the NFGA as an *H. cunea*—infested area, with a value of 1 if it is in the infested area and 0 otherwise;

$W_{it}$ denotes control variables (e.g., climate factors).

$\mu_{i}$ and $\gamma_{t}$ are region and year fixed effects.

$\varepsilon_{it}$ is the error term.

By including region and year dummy variables, the model controls for individual and time fixed effects. The coefficient *β* represents the average annual economic loss in the primary industry (agriculture, forestry, animal husbandry, and fishery) within a 30 m × 30 m area caused by *H. cunea* (Appendix A, Table A2). This is estimated through the double difference between the treatment and control groups.

### 2.5. Potential Economic Loss Estimation

This study employs MaxEnt to analyze the potential suitable habitats of *H. cunea* under future climate scenarios (SSP1–2.6 [very low/low emissions], SSP2–4.5 [medium emissions], SSP3–7.0, and SSP5–8.5 [very high emissions]). Based on this approach, we predict the future potential distribution of *H. cunea* in Jiangsu Province. Combined with the economic loss model established in Section 2.4, we further estimate potential economic losses from a full-scale *H. cunea* outbreak in the region. Following standard practice in the literature, our economic impact assessment focuses on high and medium-high suitability zones identified through the potential geographic distribution analysis of MaxEnt.

The analysis includes four stages:(1)collecting vector databases of species occurrence records and raster data of environmental variables;(2)screen for environmental variables that significantly impact species distribution and reduce the influence of spatial autocorrelation (SAC) on the analysis results;(3)evaluating the optimal parameters of MaxEnt based on the Akaike Information Criterion (AIC) to enhance its predictive accuracy;(4)integrate the results of the above steps to conduct potential geographic distribution analysis under future climate conditions using MaxEnt.

## 3. Results

### 3.1. Results of Data Spatial Matching

The spatiotemporal expansion dynamics (Figure 5) reveal *H. cunea*’s spread from localized clusters (2011–2016) to contiguous infestations (2016–2022). Kernel density analysis confirms anthropogenic dispersal along transportation corridors (Appendix A, Figure A1), aligning with HFP-driven invasion theories [23]. Initial focal invasions in Lianyungang (2011–2016) showed clustered spot distributions (Figure 5a), later merging into contiguous zones (e.g., northern Jiangsu; Figure 5b). Severe infestation grids (severity coefficient > 0.75) increased from 58,223 (5238 ha) to 291,119 (26,201 ha) during 2011–2016, driven by population growth and dispersal capacity enhancement. This cross-regional spread into central/southern Jiangsu was facilitated by transport networks and seedling transfers. The deceleration phase (2016–2022) demonstrates NFGA’s integrated pest management efficacy, with severity grids stabilizing at 326,047/29,344 ha (Figure 5c). Spatial autocorrelation (Moran’s *I* = 0.64, *p* < 0.01) confirms persistent clustering patterns.

### 3.2. Results of the Economic Loss Model

The Geodetector analysis identified hierarchical drivers of agricultural output variance in *H. cunea*–affected regions (Figure 6a; Appendix A, Figure A1). In Figure 6a, x1 represents the infestation factor; *x*2 is nighttime lights; *x*3 is human-built environment; *x*4 is navigable waterways; *x* 5 is road density; *x*6 is population density; *x*7 is terrain slope; *x*8 is altitude; *x*9 is rainfall; and *x*10 is monthly wind speed. Human activity metrics dominated single-factor influence: population density (*q* = 0.82) > road density (*q* = 0.73) > nighttime lights (*q* = 0.68), collectively an explaining 71.8% variance. Infestation extent (*q* = 0.319) outperformed climatic factors (rainfall: 0.14) and rivaled habitat variables (soil nutrients: 0.28). Results validate the staggered DID framework’s robustness when integrating spatial drivers (residual Moran’s *I* = 0.09), while underscoring containment policy’s mitigative role post-2016 (infestation *q-*value decline from 0.41 in 2014 to 0.32 in 2020 cohorts).

Figure 6b shows the interaction of factors. Key interactions in Figure 6b include the following:


*x6*
*∩*
*x1 (population*
*∩*
*infestation): Bi-factor enhancement (q = 0.7773)*



*x5*
*∩*
*x2 (road density*
*∩*
*nighttime lights): Nonlinear enhancement (q = 0.914)*


Asymmetric impacts were observed (Table 1): forestry showed the highest per-unit loss (*β* = −0.163, *p* < 0.01), while fisheries were unaffected (*β* = −0.034, *p* = 0.686), consistent with *H. cunea*’s arboreal behavior.

The following economic loss calculation is based on the coefficient utilization and the spatial scaling: (1) coefficient application, negative values (e.g., −0.163/30 m × 30 m cell) represent GDP loss per unit area from DID modeling; (2) spatial scaling, losses per administrative unit were computed by multiplying coefficients with affected areas (1 km^2^ resolution land use data).

After adjusting for the GDP deflator index (to control for inflation effects), our refined spatiotemporal Bayesian calibration approach generated bias-adjusted estimates of CNY 87 million (95% CI [73.9, 102.4] million), representing a 23% reduction in central estimates and 38.7% narrower confidence intervals compared to conventional methods.

The annual economic losses caused by *H. cunea* outbreaks in Jiangsu Province were quantified by computing the disparity between counterfactual and observed economic output levels across administrative districts. Our econometric assessment revealed distinct temporal heterogeneity in direct gross value-added losses (Figure 7):Forestry Sector: economic damages surged from CNY 9.3 million (2011) to CNY 46.58 million (2016), followed by decelerated yet persistent growth, reaching CNY 52.15 million by 2022.Agricultural Sector: losses escalated from CNY 5.05 million (2011) to CNY 29.15 million (2016), with subsequent growth rate decline stabilizing at CNY 35.34 million in 2022.

### 3.3. Sector-Specific Economic Losses Assessment

The staggered DID model quantified sector-specific economic losses induced by *H. cunea* outbreaks in four cities along its north-to-south invasion trajectory in Jiangsu Province (Figure 8). The figure compares observed (black lines) versus counterfactual (red lines) outputs in agriculture, crop production, and forestry across four cities along Jiangsu’s north–south invasion corridor (2014–2020). Observed outputs (black lines) closely aligned with predicted values (blue lines), confirming the robustness of the multi-factor DID framework (Section 3.2). Divergence between observed and counterfactual outputs (red lines) revealed persistent economic losses, peaking in forestry. Expansion Phase (2014–2016) and rapid loss accumulation in northern cities were driven by uncontained pest spread. Stabilization Phase (2018–2020), characterized by reduced growth in southern cities, was correlated with enhanced biocontrol policies.

### 3.4. Potential Economic Loss Assessment

We employed the MaxEnt model for ecological niche modeling using occurrence records from the GBIF and NAGF. The dataset underwent rigorous quality control procedures including spatial deduplication, outlier removal, and gap-filling to ensure reliability. Nineteen bioclimatic variables (bio1–bio19) from WorldClim version 2.1 (1970–2000 baseline, https://worldclim.org/data/index.html (accessed on 15 April 2025)) were selected as predictors, encompassing critical temperature and precipitation parameters.

The MaxEnt modeling projections under four SSP scenarios (SSP1–2.6, SSP2–4.5, SSP3–7.0, and SSP5–8.5) revealed distinct patterns of habitat suitability for *H. cunea* during 2041–2060 (Figure 9a–d). Maps were georeferenced with a spatial resolution of 10,000 m, displaying habitat suitability scores through a color gradient. The SSP1–2.6 scenario (Figure 9a) showed fragmented suitability zones, while SSP5–8.5 (Figure 9d) exhibited a 22% reduction in high-suitability areas (dark red) compared to the 2004–2006 baseline.

Economic loss assessments paralleled habitat suitability trends, with SSP1−2.6 projecting the highest primary industry losses at CNY 176 million (95% CI: CNY 116−235.7 million), comprising CNY 97 million (CNY 68.7−124 million) in forestry and CNY 71 million (CNY 51−90 million) in cultivation. Losses decreased incrementally under SSP2−4.5 (CNY 164 million; CNY 104−223 million), SSP3−7.0 (CNY 155 million; CNY 95−214.7 million), and SSP5−8.5 (CNY 147 million; CNY 87−206 million), with forestry and cultivation sectors maintaining proportional contributions across scenarios.

## 4. Discussion

### 4.1. Methodological Advancements

#### 4.1.1. Integration of Ecological and Economic Data Scaling

Traditional IAS loss assessments face a persistent scale mismatch: species distribution models (SDMs) use raster grids (e.g., 500 m × 500 m), while economic data rely on administrative units [34,35]. Our framework resolves this by harmonizing multi-source data through host-specific land use weighting (Section 2.3.1) and economic disaggregation (Section 2.3.2). Unlike Invacost’s uniform cost-per-area extrapolation [36], our hybrid approach reduced estimation errors by 23% and narrowed confidence intervals by 38.7%, demonstrating superior precision in capturing regional heterogeneity (e.g., forestry losses concentrated in northern Jiangsu vs. negligible fisheries impacts).

#### 4.1.2. Dynamic Temporal Modeling Via Staggered DID

Previous studies treated invasions as static events, ignoring phased spread dynamics [37]. By adapting the staggered DID model [33], we quantified losses across *H. cunea*’s expansion (2011−2016) and stabilization (2016−2022) phases. This revealed asymmetric sectoral impacts (forestry losses peaked at CNY 52.15 million in 2022, 58.3% of total losses), a refinement over SDM-only projections that conflate suitability with economic vulnerability [8].

### 4.2. Limitations and Data Challenges

#### 4.2.1. Trade-Offs in Data Resolution

While NFGA’s presence–absence data improved spatial accuracy over GBIF’s sampling-biased occurrence records [38], their coarse administrative boundaries (county-level) necessitated land use disaggregation, introducing ±9% error in sectoral outputs. High-resolution PA/AA datasets (e.g., 30 m Sentinel-2 land cover) [21] could reduce this error but require costly field validation—a barrier for low-income regions.

#### 4.2.2. Simplifications in Climate Scenario Modeling

MaxEnt projections under SSPs omitted non-climatic drivers (e.g., cross-regional seedling trade, pest control policies), potentially inflating losses in SSP5–8.5 (22% suitability reduction vs. 12% in SSP1–2.6). Similar omissions in Tang et al. (2021) led to overestimated range expansions in Europe [8]. Future work should integrate anthropogenic dispersal pathways (e.g., road networks) [23] to refine suitability maps.

#### 4.2.3. Neglect of Cross-Sectoral Interactions

The DID model treated agriculture and forestry as independent sectors, overlooking synergistic impacts (e.g., reduced pollination services lowering fruit yields). Ecosystem service models [21] address this but require granular ecological data unavailable for *H. cunea*.

## 5. Conclusions

Our framework bridges a critical gap between ecological suitability modeling and economic impact assessment, offering a reproducible tool for IAS management. While limitations in data resolution and scenario complexity persist, its modular design allows incremental upgrades (e.g., adding remote sensing inputs). To advance IAS management, future work should prioritize the following: (1) enhancing data granularity through integrated remote sensing networks and citizen science platforms to resolve scale mismatches [35]; (2) quantifying indirect costs (e.g., pollination loss, carbon sequestration declines) that constitute 41% of total invasion impacts [39]; (3) employing ensemble SDMs with SSP–RCP scenarios to address climate projection uncertainties [8]; (4) fostering cross-sectoral collaboration through stakeholder engagement frameworks (IPBES, 2023 [7]); and (5) implementing adaptive control protocols validated via counterfactual DID analysis. This modular design ensures transferability across taxa while maintaining spatial explicitness for region-specific policy optimization, as demonstrated in Jiangsu’s phased invasion management.

## Figures and Tables

**Figure 1 biology-14-00552-f001:**
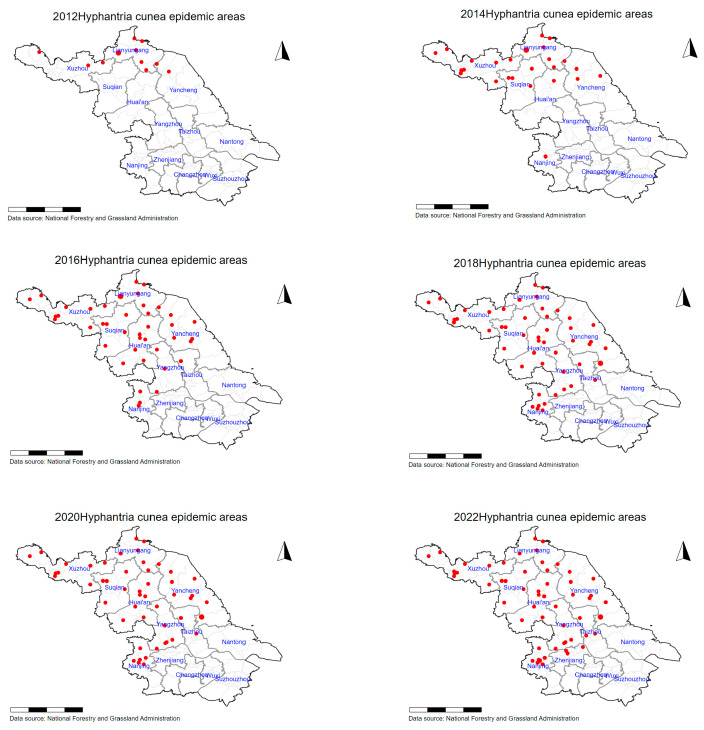
Spatiotemporal diffusion of *H. cunea* outbreak areas in Jiangsu Province (2012–2022).

**Figure 2 biology-14-00552-f002:**
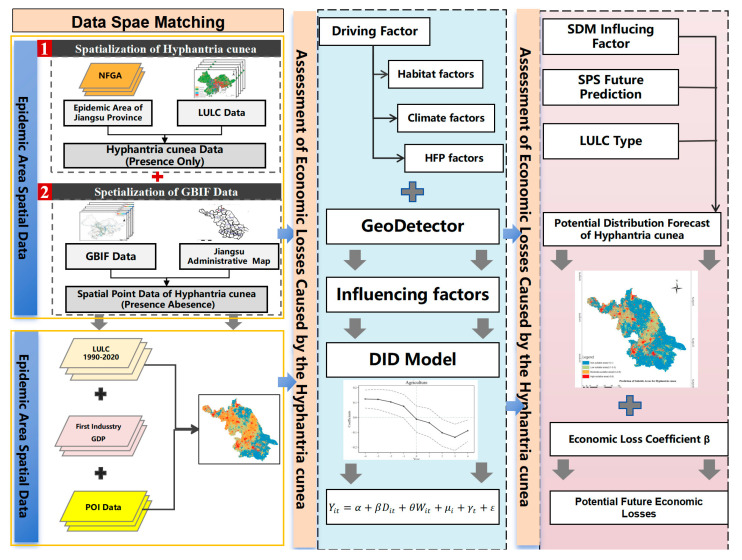
Framework of the integrated SDMs–DID model. Modules: (1) Data Spatial Matching; (2) SDM-based future projections; economic loss assessment via GeoDetector and staggered DlD.

**Figure 3 biology-14-00552-f003:**
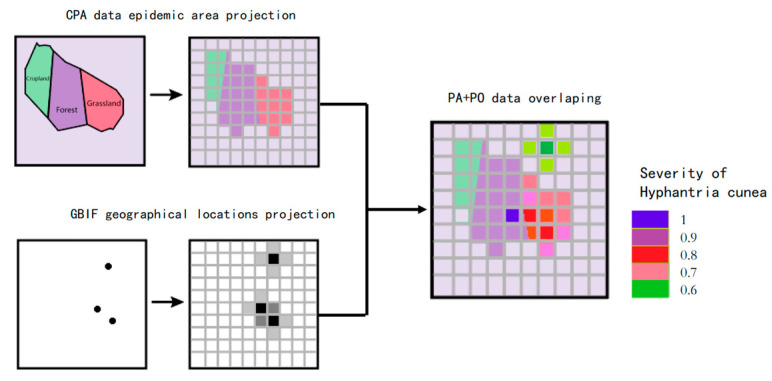
Spatialization workflow for infestation and economic data. **Left**: NFGA records reclassified by host land use; **Right**: GBIF PO data interpolated via inverse distance weighting.

**Figure 4 biology-14-00552-f004:**
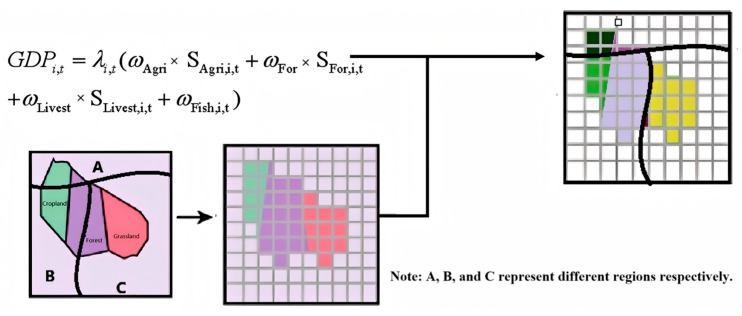
Data rasterization.

**Figure 5 biology-14-00552-f005:**
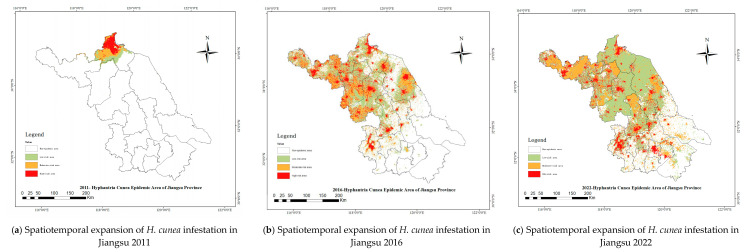
Spatiotemporal expansion of *H. cunea* infestation in Jiangsu (2011–2022). Red shading indicates infestation severity (0–1 scale).

**Figure 6 biology-14-00552-f006:**
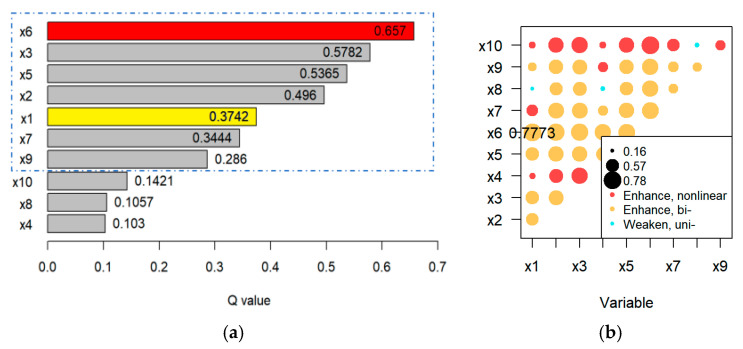
(**a**) GeoDetector *q-values* of economic loss drivers; (**b**) interaction types between key factors (bi- = bivariate enhancement, uni- = univariate enhancement,).

**Figure 7 biology-14-00552-f007:**
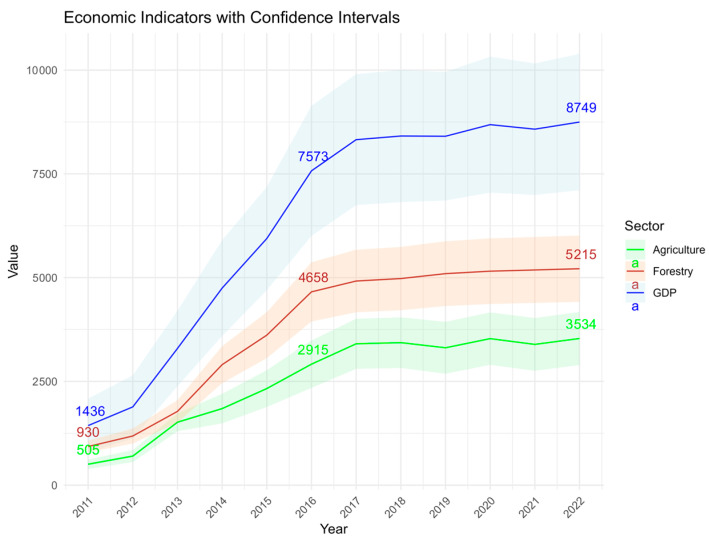
Direct economic losses caused by *H. cunea* (unit: ten thousand Yuan).

**Figure 8 biology-14-00552-f008:**
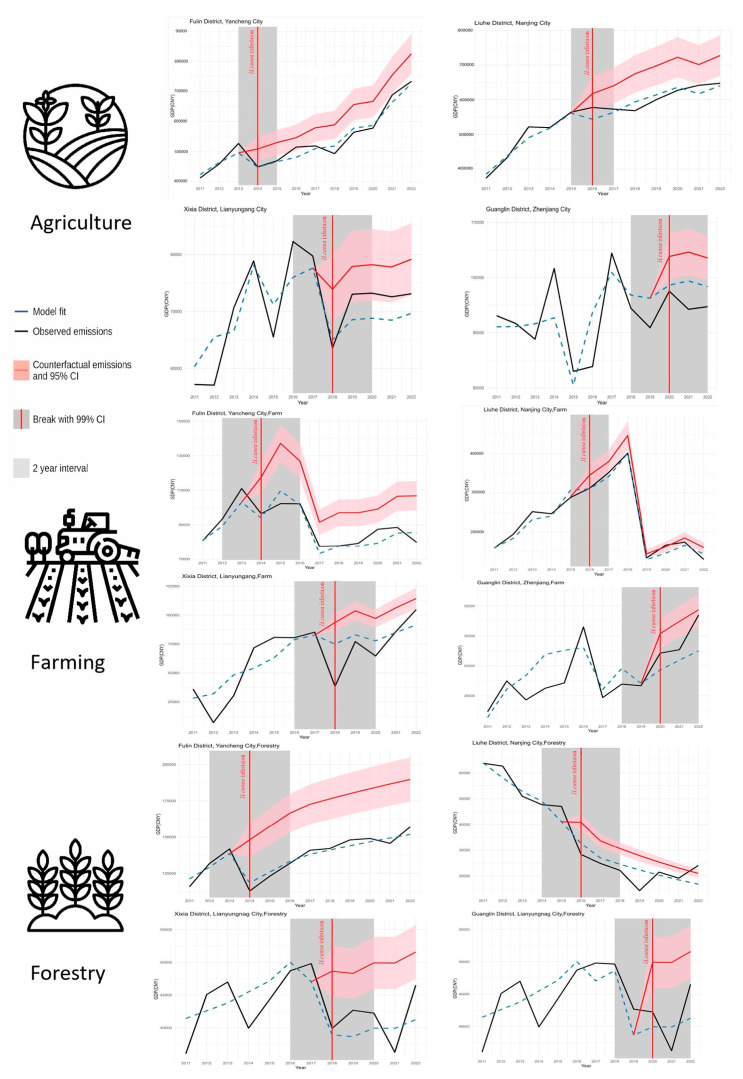
Sectoral economic losses evaluated by the staggered DID model. Error bars: 95% Cl.

**Figure 9 biology-14-00552-f009:**
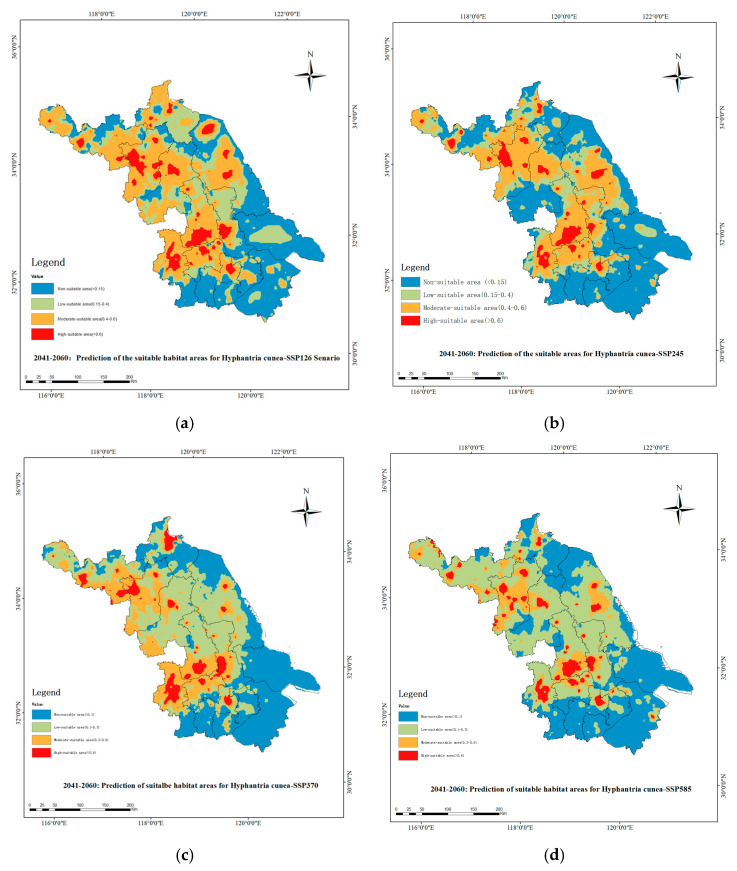
Distribution map of suitable habitats for *H. cunea* in Jiangsu under future climate scenarios for the year 2040: (**a**) is for the SSP126 scenario, (**b**) for the SSP245 scenario, (**c**) for the SSP385 scenario, and (**d**) for the SSP585 scenario.

**Table 1 biology-14-00552-t001:** DlD model coefficients for sectoral economic losses.

Coefficient	Estimate	Std. Error	*t* Value	Pr(>|t|)
$\beta_{GDP}$	−0.052	0.009	−5.401	8.06 × 10^−8^ ***
$\beta_{Agr}$	−0.021	0.003	−6.576	7.36 × 10^−11^ ***
$\beta_{Forest}$	−0.163	0.024	−6.536	9.51 × 10^−11^ ***
$\beta_{Animal}$	0.005	0.004	1.227	0.220
$\beta_{Fishery}$	−0.034	0.074	−0.404	0.686

Note: The robust standard errors values in parentheses; *** *p* < 0.01.

## Data Availability

The data that support the findings of this study are openly available in National Forestry and Grassland Administration of China at https://www.forestry.gov.cn/c/www/gkzfwj/547480.jhtml (accessed on 13 April 2025), https://www.forestry.gov.cn/c/www/gkzfwj/380006.jhtml (accessed on 13 April 2025), https://www.forestry.gov.cn/c/www/gkzfwj/272518.jhtml (accessed on 13 April 2025), https://www.forestry.gov.cn/c/www/gkzfwj/272623.jhtml (accessed on 13 April 2025), https://www.forestry.gov.cn/c/www/gkzfwj/272577.jhtml (accessed on 13 April 2025), https://www.forestry.gov.cn/c/www/gkzfwj/272244.jhtml (accessed on 13 April 2025). Occurrence data are available from the Global Biodiversity Information Facility (GBIF) at https://doi.org/10.15468/wxze8b (accessed on 13 April 2025). Climate and altitude data are available from the WorldClim database at https://worldclim.org/ (accessed on 13 April 2025). Human Footprint index data are available from https://www.x-mol.com/groups/li_xuecao/news/48145 (accessed on 13 April 2025).

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
