# Peer review of "Research on the Economic Loss Model of Invasive Alien Species Based on Multidimensional Data Spatialization—A Case Study of Economic Losses Caused by Hyphantria cunea in Jiangsu Province"

_biology, 2025, doi:10.3390/biology14050552_

Round 1
Reviewer 1 Report (Previous Reviewer 3)
Comments and Suggestions for Authors
The new version of the article is well written and very clear.
Author Response
Thank you for your professional advice and recognition. We've carefully checked that part to ensure language quality.
Reviewer 2 Report (Previous Reviewer 2)
Comments and Suggestions for Authors
Dear Authors, please see feedback below:
Abstract
1. Explain methodological innovation: Give a brief explanation of the DID model, GeoDetector, and MaxEnt's use, as well as how they work in tandem to increase the accuracy of economic loss assessment.
2. Highlight useful uses of the model for resource allocation, policymaking, and managing invasive species in light of future climatic scenarios to bolster the conclusion.
Introduction
1. Divide the lengthy paragraph into two sections to make it easier to read and highlight important ideas. The first section should address the worldwide problem of biological invasions, followed by a section that focuses on the particular effects of Hyphantria cunea.
2. Give a more thorough description of how your strategy differs from conventional approaches, emphasizing the Difference-in-Differences model, the usage of GeoDetector, and the integration of spatialized economic information.
3. Highlight how your research might be used in real-world situations by clearly connecting the anticipated economic effects to concrete solutions like focused initiatives or altered laws.
Methods
1. Clearly distinguish between the initial stages (2011-2016) and later stages (post-2016) of H. cunea spread. This will help readers understand the progression of the invasion and its increasing economic impact over time.
2. The framework you've developed for assessing both current and potential economic losses due to H. cunea is well-structured and innovative- very well descrived.
Results
1. The study tracks the spread and economic impact of Hyphantria cunea in Jiangsu, China, from 2011 to 2022. Using a Difference-in-Differences model, it estimates economic losses of 87 million CNY. Key drivers include population density and road networks, with forestry and agriculture sectors facing significant damage over time- The results are interesting and well presented.
Conclusion
1. To enhance IAS management, future work should focus on improving data quality, integrating remote sensing and real-time monitoring, expanding to indirect costs, adopting ensemble SDMs to address climate uncertainty, collaborating with stakeholders, and evaluating control measures, while ensuring the framework remains adaptable to other species and socioecological contexts.
Minor English edits required.
Author Response
Please see the attachment

This manuscript is a resubmission of an earlier submission. The following is a list of the peer review reports and author responses from that submission.
Round 1
Reviewer 1 Report
Comments and Suggestions for Authors
Dear Dr. Ettore Randi, Editor-in-Chief of the section of Conservation Biology and Biodiversity in Biology.
Thank you for inviting me to review the paper named “Research on the Economic Loss Model of Invasive Alien Species Based on Multidimensional Data Spatialization - A Case Study of Economic Losses Caused by Hyphantria cunea in Jiangsu Province”. In general, the paper is well written, with no language errors. However, I believe the authors are not sure what their paper is about. If it is about the impacts of alien species they have to take more time in the introduction talking about cases of alien species and what they can cause, in China and elsewhere. If it is a paper about methodology only, they should focus on the comparison between different models and their results, to support the new methodology they are proposing. Moreover, the results reported do not align with the model outputs. For instance, the results demonstrate a significant negative effect of the variables, but the authors directly quantify this as monetary losses, which is not what the output of the models suggest. These descriptions must be revised to reflect the actual results accurately. Similarly, other results segments lack transparency concerning data sources, estimations and p-values. Finally, the discussion is extremely short. Despite the authors claiming that their results are a novelty, they do not compare them with any other study. The statistical evidence and context must be provided to substantiate these claims. In the end, no scientific progress was made in this paper. It ends up being a mix between a revision, some random and wrongly interpreted results and no discussion about its real and future implications.
To be accepted for publication, this paper has to be rebuilt from the ground up. It needs a very specific objective, again being methodological or about species invasion impact, to be able to move science forward. I believe that the authors have good data and a good new methodology to be applied. However, they lack a focus. As soon as this focus is reached, I believe that this research can be a very good advance for science.
Thank you for considering these recommendations.
Major:
Introduction:
Line 86: Add the citation of @risk.
Lines 103 to 106: References for DID models?
Line 113 to 106: References for por quality?
Lines 81 to 128: All of this is Methodology and should be moved to the proper section.
Line 129 to 133: As said in the previous comment, you spend too much time of your introduction talking about methodology. At the end, you only added a feel lines about the species you are interested in.
Methods:
Lines 156 to 158: This justification should be at the end of the introduction, not in the methods.
Line 195: Reference for the National Forestry and Grassland Administration?
Lines 295 and 296. You are basically repeating the last sentence in the previous paragraph.
Lines 297 to 299: These are objectives. It should be presented at the end of the introduction.
Results:
Lines 376 to 379: These are good statements for the beginning of the discussion, not the Results section. Please, remove it. You can start your paragraph directly in “The DID model” (line 379).
Lines 399 to 418: This entire section is hard to understand if the information here is new, brought by your results, or you are informing something that is already known. Please, if it is your results, rephrase and restructure the paragraph to make this sure. If this information is not new, please, move it to the proper sections, like the Introduction or Discussion.
Lines 459 to 468: The results and text presented here are not what the models are actually showing. For instance, what Table 2 actually presents is that “GDP” has a significant negative effect on the Economic Loss Coefficient. The same can be said by the “Agr” and “Forest”, both also negatively significant. The model does not present the amount of yuan lost. They only present the effect of the predictor variable on the response variable.
Line 470 to 487: It is impossible to know where this data is coming from. The authors don’t show the statistics and p-values associated with these estimations of loss and gain. Moreover, as previously pointed, it is hard to understand if this information is new for this paper or if it is already known. Hence, the correlation and causation presented in this paragraph seems to come out of the blue. Without any support.
Lines 490 to: This is Methodology. Please, move it to the proper section.
Lines 508 532: This time, the authors do not present any of the estimations for the models. Hence, it is impossible to know where these economic losses are coming from. Please, provide the estimation of the models and also their p-values to support your estimates of economic losses and gains.
Figure 5: It is impossible to read the legends and the axis of the figure. Please, provide a figure with a better resolution.
Figure 6: It is impossible to read the legends and the axis of the figure. Please, provide a figure with a better resolution.
Figure 7: It is impossible to read the legends and the axis of the figure. Please, provide a figure with a better resolution.
Figure 8: It is impossible to read the legends and the axis of the figure. Please, provide a figure with a better resolution.
Figure 9: It is impossible to read the legends and the axis of the figure. Please, provide a figure with a better resolution.
Discussion:
Lines 534 to 539: This is not and discussion, this is an introduction. The authors are not comparing their results with results found in another place.
Lines 541 to 558: This discussion is very small, and actually do not make any science progress. The authors did not bring any other bibliography to compare and discuss with their results. Besides that, since part of the results are not complete or wrongly interpreted, they are not able to support the claims presented here in the discussion.
Appendix:
Figures: It is impossible to read the legends and the axis of the figure. Please, provide a figure with a better resolution.
Minor:
Simple summary:
Line 14: Please, right in the beginning, spell the whole acronym.
Line 17: What is DID? Again, please, be careful with the acronyms right in the beginning.
Line 16 and 17: This sentence, “Our study integrated data and used DID and GeoDetector models.” it is not important here, you can remove it.
Abstract:
Line 26: Species name is not in italic.
Line 32: Species name is not in italic.
Line 33: Species name is not in italic.
Keywords:
Line 43: Species name is not in italic. Besides, most of these words are in the abstract or title. Please, choose other synonyms. It does not make sense to be the same words that you used before.
Introduction:
Line 77: Look for uppercase in “application”.
Line 111 and 112: The phrase “are generally of higher quality” is repeated.
Methods:
Lines 137 and 138: Species name is not in italic.
Line 141: Species name is not in italic.
Lines 145 and 149: Species name is not in italic.
Line 152 and 155: Species name is not in italic.
Fig 1 and line 161: Species name is not in italic.
Line 174: Species name is not in italic.
Line 189: Species name is not in italic.
Line 194: Species name is not in italic.
Line 198: Species name is not in italic.
Line 202: Species name is not in italic.
Line 207: Species name is not in italic.
Line 210: Species name is not in italic.
Line 212: Species name is not in italic.
Line 214: Species name is not in italic.
Lines 220 and 222 and 224 and 226: Species name is not in italic.
Lines 231 and 232: Species name is not in italic.
Line 262: Species name is not in italic.
Line 286: Species name is not in italic.
Line 291: Species name is not in italic.
Lines 335 and 341: Species name is not in italic.
Line 345: Species name is not in italic.
Line 349: be careful of the space in “The causal”.
Line 353: Species name is not in italic.
Line 363: Species name is not in italic.
Results:
Lines 380 and 383 and 384: Species name is not in italic.
Lines 400 and 405: Species name is not in italic.
Line 459: Species name is not in italic.
Lines 470 and 472: Species name is not in italic.
Line 493: Species name is not in italic.
Figure 5: Species name is not in italic.
Figure 6: Species name is not in italic.
Discussion:
Line 553: Species name is not in italic.
Reviewer 2 Report
Comments and Suggestions for Authors
Dear Authors,
Please see the comments below:
1. Lack of background on the fall webworm in the introduction.
2. Please ensure that species names are in italics.
3. Please include some statistics in lines 29-36 to support the information.
4. Include a few ways that the study has addressed the challenges of IAS- line 41.
5. Expand on IAS- start broad and then be specific on the type eg. Plants or animals.
6. The information regarding the IAS Hyphantria cunea is vague and must be improved since it is a major aspect of the study. Why was the species chosen?
7. Please include a range of coordinates in the research area if possible,
8. Figures 1-4 display the maps well, however, the maps in Figure 5 are difficult to view- kindly enlarge them.
9. Enlarge Figure 6
10. Figures 8 and 9 are of very poor quality.
11. Improve language- line 543.
12. Discussion is lacking an in-depth comparison with other studies. Please improve.
13. Please include the conclusion section.
Comments on the Quality of English LanguageMinor English editing required.
Reviewer 3 Report
Comments and Suggestions for Authors
The article is innovative in terms of methodology for analyzing economic loss data caused by an insect in a region of China. Several aspects deserve to be highlighted:
- "Hyphantria cunea/ H. cunea" needs to appear in italics throughout the text;
- the first citation of this binomial must be accompanied by the name of the descriptor and year (Drury, 1773)
- Keywords: add "MaxEnt" and "future climatic conditions";
- line 38: "more accurate" than....;
- line 77: . ? Application...?
- lines 137-139: better in the Introduction?
- lines 156-158: better in the Discussion?
- Figure 2: not all abbreviations are properly explained, or were explained much later in the text;
- line 221: "control points": it is necessary to explain better;
- Figures 3 - 6: without a traditional call in the text. All with many explanations in the caption. Some of this should be in the body of the text;
- line 262: formulas?
- lines 266-267: what is the meaning of "S"?
- lines 302 and 344: start with "where";
- line 306: q=0 XXX?
- line 310: detector... detects; demonstrates?
- line 322: "natural experiments" x economics?
- line 347: one of the terms does not match those in the equation and the explanation of two others is missing;
- line 415: "controlled" or "with less increase"?
- line 444: further evaluation: where and when?
- Figure 9 with axis captions not visible;
- Discussion without references cited! And the comparison with other articles?
Round 2
Reviewer 1 Report
Comments and Suggestions for Authors
Dear Dr. Ettore Randi, Editor-in-Chief of the section of Conservation Biology and Biodiversity in Biology.
Thank you for inviting me to review the paper named “Research on the Economic Loss Model of Invasive Alien Species Based on Multidimensional Data Spatialization- A Case Study of Economic Losses Caused by Hyphantria cunea in Jiangsu Province”. In general, the paper keeps to be well written. However, the authors did not address many of my concerns and questions from the previous versions. Besides that, there is still some confusion regarding which part of the text should be in its proper section: methods in introduction, methods in results, results in discussion. Unfortunately, I believe that this paper needs more time to be able to be published. Hence, I reject it.
Thank you for considering these recommendations.
Sincerely,
Major:
Simple summary:
Line 14: Use “Invasive alien species (IAS)”, as you used in the abstract.
Abstract:
Lines 37 to 39: This is not results, this is methods.
Introduction:
Line 65 to 73: Where are your hypotheses and predictions?
Methods:
Lines 105 to 116: Most of this information were already present in the introduction. You can’t repeat it here. Or be more general in the introduction and present the focus species in the methods, or remove this part.
Results:
Trough results: Remove the “Results of” from the title of your sessions.
Lines 376 to 379: These are good statements for the beginning of the discussion, not the Results section. Please, remove it. You can start your paragraph directly in “The DID model” (line 379).
Lines 399 to 418: This entire section is hard to understand if the information here is new, brought by your results, or you are informing something that is already known. Please, if it is your results, rephrase and restructure the paragraph to make this sure. If this information is not new, please, move it to the proper sections, like the Introduction or Discussion.
Lines 459 to 468: The results and text presented here are not what the models are actually showing. For instance, what Table 2 actually presents is that “GDP” has a significant negative effect on the Economic Loss Coefficient. The same can be said by the “Agr” and “Forest”, both also negatively significant. The model does not present the amount of yuan lost. They only present the effect of the predictor variable on the response variable.
Line 470 to 487: It is impossible to know where this data is coming from. The authors don’t show the statistics and p-values associated with these estimations of loss and gain. Moreover, as previously pointed, it is hard to understand if this information is new for this paper or if it is already known. Hence, the correlation and causation presented in this paragraph seems to come out of the blue. Without any support.
Lines 490 to: This is Methodology. Please, move it to the proper section.
Lines 508 532: This time, the authors do not present any of the estimations for the models. Hence, it is impossible to know where these economic losses are coming from. Please, provide the estimation of the models and also their p-values to support your estimates of economic losses and gains.
Figure 5 to 9: It is hard to read the legends and the axis of the figure. Please, provide a figure with a better resolution.
Discussion:
Lines 440 to 455: This is not discussion, this is methods.
Through Discussion: The discussion still present itself as a “no” discussion. Authors do not bring any other author to compare and properly discuss their results. Hence, most of what is present in the discussion is just results.
Conclusion:
Lines 487 to 492: This is not discussion, this is introduction mixed with methods.
